# Metal Bi Loaded Bi$_2$Ti$_2$O$_7$/CaTiO$_3$ for Enhanced Photocatalytic Efficiency for NO Removal under Visible Light

**Diyuan Du** [1], **Menglin Shi** [1], **Qingqing Guo** [1], **Yanqin Zhang** [1], **Ahmed A. Allam** [2], **Ahmed Rady** [3] **and Chuanyi Wang** [1,*]

[1] School of Environmental Science and Engineering, Shaanxi University of Science and Technology, Xi'an 710021, China; dudiyuan2021@163.com (D.D.); menglinshi0112@163.com (M.S.); guoqingqing00544@163.com (Q.G.); zhangyanqin1211@163.com (Y.Z.)

[2] Zoology Department, Faculty of Science, Beni-Suef University, Beni-Suef 62511, Egypt; ahmed.aliahmed@science.bsu.edu.eg

[3] Department of Zoology, College of Science, King Saud University, P.O. Box 2455, Riyadh 11451, Saudi Arabia; ahabdo@ksu.edu.sa

[*] Correspondence: wangchuanyi@sust.edu.cn

**Abstract:** NO has caused many serious environmental problems and even seriously threatened human health. The development of a cheap and efficient method to remove NO from the air has become an urgent need. In this paper, a novel nanocomposite metal-semiconductor photocatalyst Bi-Bi$_2$Ti$_2$O$_7$/CaTiO$_3$ was prepared. Compared to the original Bi$_2$Ti$_2$O$_7$/CaTiO$_3$, the modification by the metal Bi increased its photocatalytic activity from 25% to 64% under visible light irradiation. The improved photoactivity owns to the SPR effect and the electron capture effect of Bi metals in metal-semiconductor loaded systems improving the separation efficiency of electron-hole pairs and significantly improving the light absorption capacity of the composite photocatalyst. The capture experiment of active species showed that •OH, •O$_2^-$, h$^+$ and e$^-$ are the main active species in the photocatalytic conversion of NO. This work provides new insights into the conformational relationships of Ti-based photocatalysts for NO removal.

**Keywords:** NO removal; photocatalysis; Bi; metal-semiconductor photocatalyst





## 1. Introduction

With the increase in human productivity and the rapid development of the world's economy, energy consumption has increased dramatically in recent decades, leading to a significant increase in nitrogen oxide concentrations in the atmosphere [1,2]. For example, acid rain, the greenhouse effect and photochemical smog can cause great harm to the environment and even threaten human health [3,4]. Traditional pollutant removal technologies such as chemisorption, wet oxidation and selective decomposition–reduction with thermal catalysts require high temperatures and reducing agents, making it economically unfeasible to remove nitrogen oxides from the air at parts per billion (ppb) levels [5,6].

Photocatalysis is a technology that uses semiconductors as photocatalysts to degrade pollutants through redox reactions by using sunlight as a driving force to generate electron-hole pairs under illumination [7]. It is widely used in treating water and air pollution remediation because of its low energy consumption, low cost and environmental friendliness [8]. Since the discovery that the N-type semiconductor electrode TiO$_2$ can decompose water under UV light to produce H$_2$ and O$_2$, many efforts have been directed to semiconductor-based photocatalysis. TiO$_2$ has been more often studied by researchers due to its high reactivity and chemical stability under UV light [9].

Among a large number of catalytic materials widely known today, metal oxides in the form of ABO$_3$ have been studied by many researchers for their great potential in environmental applications due to many advantages, such as structural flexibility, low



price and photothermal stability [10]. $CaTiO_3$ is a highly photoactive material, but it has the disadvantages of a large band-gap width and low quantum efficiency, which leads to easy compounding of electron-hole pairs and is only responsive to UV light, which limits its overall photocatalytic performance [11,12]. So far, it has been modified in many ways, including element doping [13], metal deposition [14], defect construction and heterojunction formation. Jiang et al. [15], constructed a new heterostructure of the Z-type $MoS_2/CaTiO_3$ by means of a morphological control strategy. The heterogeneous structure of the Z-type $MoS_2/CaTiO_3$ has more effective charge carrier separation efficiency, prolongs the lifetime of photogenerated charges and improves the photocatalytic activity of the modified composite. A novel all-solid Z-shaped heterojunction for $CaTiO_3/Cu/TiO_2$ was designed by Yang et al. [16]; it is because of the construction of the novel Z-type charge transport path of $CaTiO_3/Cu/TiO_2$ that the rapid separation of the photogenerated carriers is possible, and thus the photocatalytic efficiency of this composite is improved. Zhao et al. [17] prepared a $CaTiO_3/g-C_3N_4$ photocatalyst by the calcining method. After the modification, $CaTiO_3/g-C_3N_4$ achieved 100% removal of methylene blue (MB) and 87.7% removal of levofloxacin (LVF) within 120 min under irradiation of a 500 W mercury lamp. However, the forbidden band width of $CaTiO_3$ is 3.4 eV, and it only responds to UV light, which limits its application in practice. We hope that the modification of $CaTiO_3$ can effectively broaden the photo response range of the material and accelerate the electron and hole transfer, thus improving the photocatalytic efficiency of the material.

In this work, $Bi-Bi_2Ti_2O_7/CaTiO_3$ was synthesized by a mild in situ hydrothermal method. The introduction of Bi nanospheres gave rise to the SPR effect, which greatly improved the separation efficiency of the electron-hole pairs, resulting in a significant increase in the photocatalytic activity of the composite. It was found that the loading of the metal Bi favors the generation of more active species on the surface of the $Bi_2Ti_2O_7/CaTiO_3$, which can improve the light absorption capacity and electron transport efficiency of the $Bi_2Ti_2O_7/CaTiO_3$ materials.

## 2. Results and Discussion

### 2.1. Phase and Composition

The phase analysis of the as-obtained products was determined by XRD (Figure 1a). The X-ray diffraction analysis of the sample shows four typical diffraction peaks for Bi. The peak at 27.2° belongs to the (012) plane, the peak at 37.9° to the (104) plane, the peak at 39.6° to the (110) plane and the peak at 48.7° to the (202) plane. All sharp peaks can be rightly matched with the orthogonal phase of Bi (JCPDS PDF No. 85-1329). The XRD patterns of $CaTiO_3$ and $Bi_2Ti_2O_7$ samples were reported previously [18]. Both synthesized $CaTiO_3$ and $Bi_2Ti_2O_7$ have four sharp and intense peaks, indicating that the synthesized samples have good crystallinity. XRD patterns confirmed that the Bi-BTO/CTO ternary nanocomposites with high crystallinity were obtained by a one-step in situ hydrothermal process without further processing. With the introduction of Bi, the characteristic peak of Bi was detected in BTO/CTO, indicating the successful synthesis of Bi-BTO/CTO [19]. The specific surface area plays an important role in the research of nanomaterials. The size of the specific surface area of the material affects the photocatalytic activity of the material to some extent. The larger the specific surface area of the material, the more active sites it can provide. The nitrogen adsorption–desorption isotherms for different mass ratios of BTO/CTO are shown in Figure 1b. According to the IUPAC classification, the isotherms for all samples are both type IV, revealing the existence of the mesoporous structure [20]. The specific surface areas of BTO/CTO, Bi-BTO/CTO-15, 25, 50 and 100 are 8.59, 8.9, 14.8, 16.2 and 23.29 $m^2/g$, respectively. The increase in the specific surface area of the Bi-BTO/CTO-X composites can be attributed to the introduction of Bi nanoparticles providing more active sites.

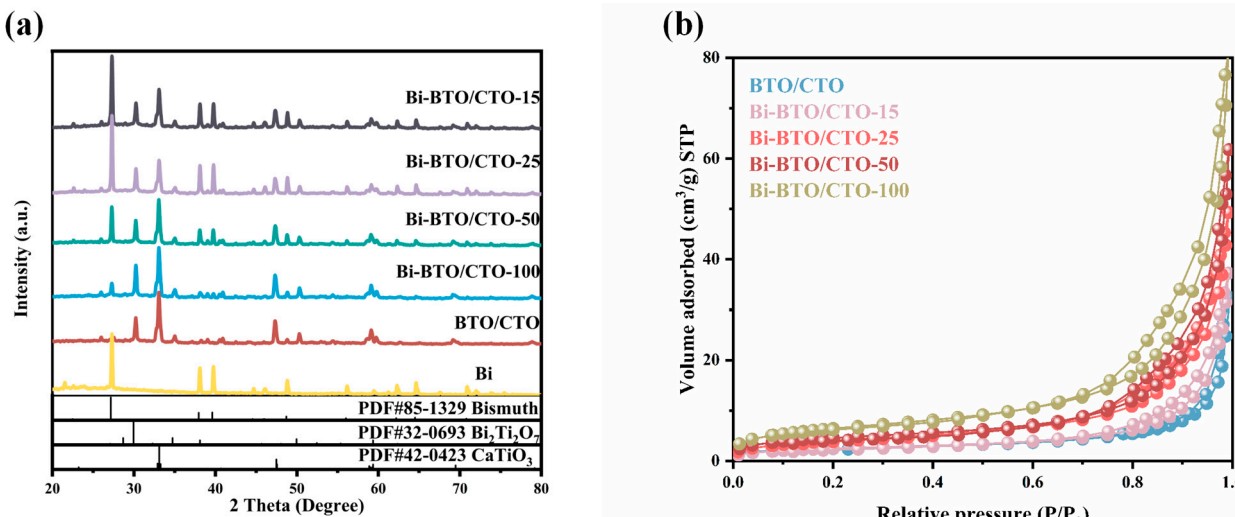

**Figure 1.** XRD patterns of different quality ratio Bi and Bi-BTO/CTO−X samples (**a**), the $N_2$ adsorption-desorption isotherms of the BTO/CTO and Bi−BTO/CTO−X samples (**b**).

The surface chemical state and elemental composition of Bi-BTO/CTO-50 were analyzed by X-ray photoelectron spectroscopy XPS (Figure 2). It is clear from the XPS spectra that Bi-BTO/CTO-50 is composed of Ca, O, Ti and Bi elements. The dashed line represents the degree of peak offset. The XPS spectrum of Bi 4f is shown in Figure 2a. The characteristic peaks of the pre-load Bi 4f XPS spectra distributed at 156.7 and 162.2 eV are generally attributed to the characteristic peaks of the Bi-Bi bond [21], while the peaks approximately 158.8 and 164.1 eV after loading are generally attributed to the typical peaks of the Bi-O bond. After surface etching, the Bi-Bi peak was significantly enhanced, indicating that only a thin bismuth oxide layer was formed on the surface of the element Bi, and the formation of the Bi-O layer could prevent further oxidation of the dioxygen layer. The XPS spectrum of O 1s is shown in Figure 2b. For pure BTO/CTO, the high-resolution spectra of O 1s can be divided into three peaks. The peak located at 529.3 eV can be attributed to the Bi-O bond, the peak observed at 530.8 eV is attributed to lattice oxygen, and the peak at 532.6 eV corresponds to surface chemisorbed oxygen from the atmosphere. For Bi-BTO/CTO-50, two characteristic peaks of O 1s distributed at 529.6 and 530.8 eV were detected, which could be explained by the lower binding energy of the Bi-O bond, so that it can assimilate active species ($H_2O$ or $O_2$) at the surface. The high-resolution spectrum of Ca 2p is shown in Figure 2c. For BTO/CTO, two peaks were observed, which are located at 346.2 eV and 349.8 eV and attributed to the Ca $2p_{3/2}$ orbital Ca $2p_{1/2}$ orbital of $Ca^{2+}$, respectively. For Bi-BTO/CTO-50, the two characteristic peaks shift towards lower binding energy, which may be due to the influx of $e^-$ into the Bi-BTO/CTO-50, causing the electron cloud density of the Bi-BTO/CTO-50 to increase [22]. The high-resolution spectrum of Ti 2p is shown in Figure 2d. For BTO/CTO, the peaks at 458.2 eV and 464.7 eV are attributed to the $2p_{3/2}$ and $2p_{1/2}$ orbitals of the $Ti^{3+}$ species [23]. For Bi-BTO/CTO-50, the two characteristic peaks are slightly shifted towards lower binding energy, which may be caused by the easy oxidation of $Ti^{3+}$ by $O_2$ in the air.

### 2.2. Morphology of the Photocatalysts

Figure 3 shows the SEM images of pure BTO/CTO and Bi-BTO/CTO-50 samples. BTO/CTO is mainly composed of numerous flake petals. CTO with a flake size of 100–200 nm is densely decorated with irregularly shaped BTO particles, which is conducive to the growth of Bi nanoparticles on the surface of BTO/CTO (Figure 3a). As shown in Figure 3b, it is clear that the Bi nanoparticles with diameters of 200–300 nm are dispersed on the BTO/CTO. A tight metal-semiconductor interface is formed in the Bi-BTO/CTO nanocomposite, which facilitates the rapid separation of carriers. Figure 3c,d shows that the

CTO is in the shape of an open petal and the BTO particles are randomly dispersed on the CTO surface with sizes ranging from 40 nm to 100 nm. The EDX analysis pattern shows that O, Bi, Ti and Ca elements are uniformly distributed in the sample, indicating the successful synthesis of Bi-BTO/CTO. The single-point scanning proves that the nano-spherical sample is elemental Bi, and the Bi-BTO/CTO-X sample is successfully synthesized.

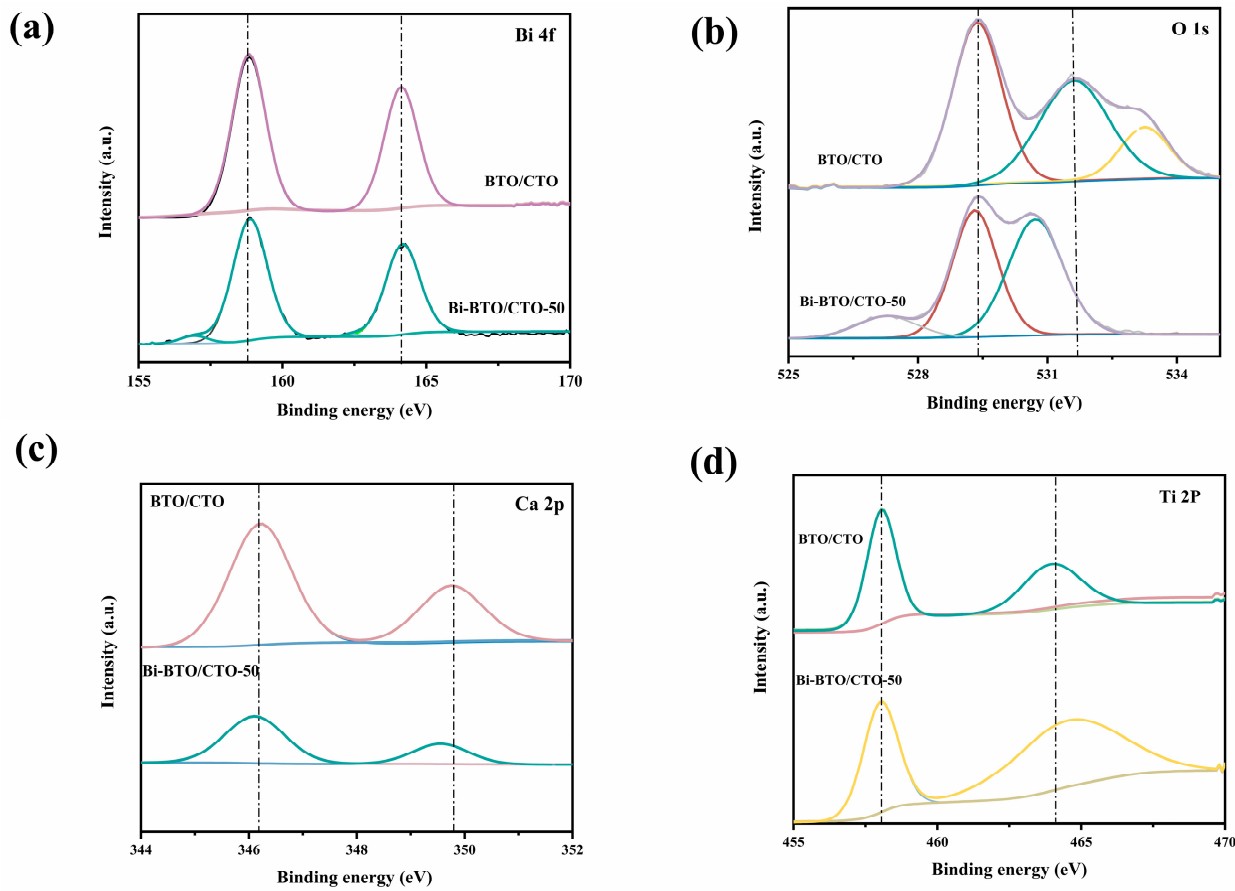

**Figure 2.** The XPS spectra of the BTO/CTO and Bi−BTO/CTO−50 sample, Bi 4f (**a**), O 1s (**b**), Ca 2p (**c**) and Ti 2p (**d**).

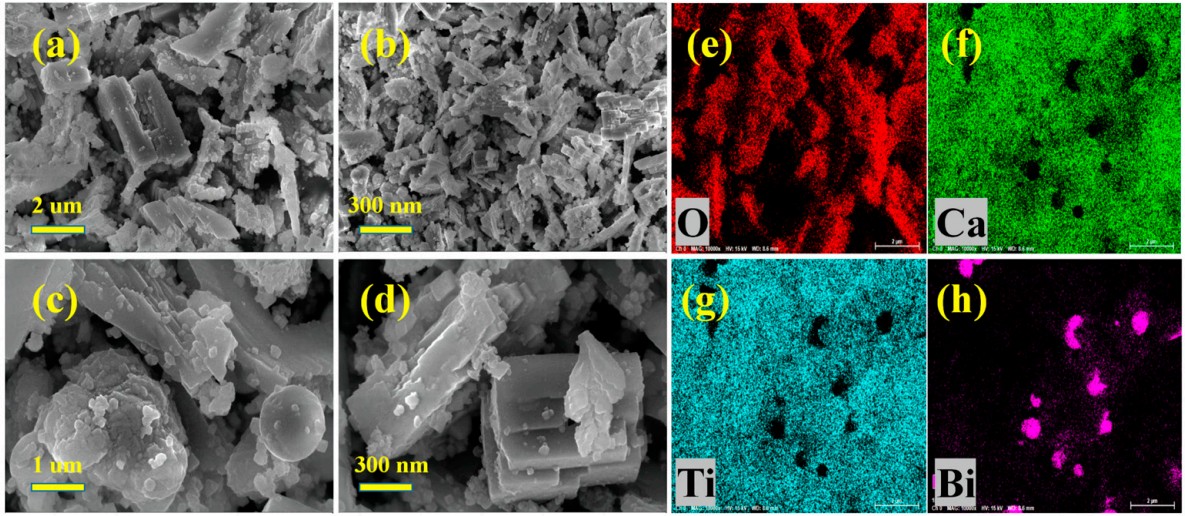

**Figure 3.** SEM images of Bi−BTO/CTO−50 (**a**,**b**), BTO/CTO (**c**,**d**), EDS elemental mapping images of Bi−BTO/CTO−50 (**e–h**).

### 2.3. Optical and Photoelectrochemical Properties

The light absorption characteristics of semiconductors have a very important influence on their photocatalytic performance and can be determined by UV-vis measurements [24]. Figure 4a displays the UV-vis absorption spectra of pristine CTO/BTO nanomaterials and Bi-BTO/CTO-X nanomaterials. The Bi-BTO/CTO-X is gradually red-shifted after loading with Bi, and the visible light absorption is significantly enhanced from 200 nm to 800 nm. Based on the finite integral technology, C. Pflaum et al. [25] simulated the local electric field generated by the metal Bi with a Maxwell solver, which significantly improved the light absorption range of the system and developed photocatalytic materials with broad spectrum absorption characteristics. As shown in Figure 4b, after the modification, the band gap of Bi-BTO/CTO-50 was narrowed to 3.14 eV, which improved the photocatalytic performance of the material to some extent. This phenomenon confirms that the loading of Bi improves the visible light absorption of BTO/CTO. The built-in electric field effect induced by the SPR effect of the metal Bi promotes not only the transfer of photogenerated electrons but also the separation and migration of photogenerated carriers [26], thus improving the photocatalytic efficiency of BTO/CTO nanocomposites.

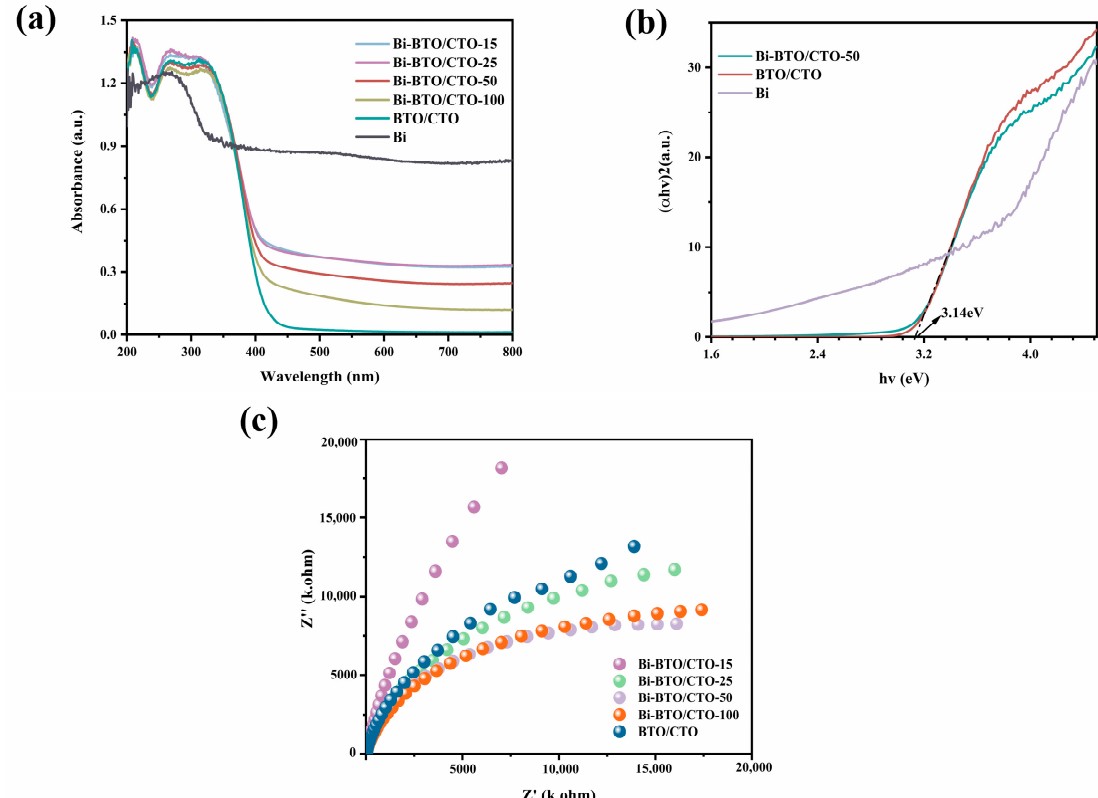

**Figure 4.** UV-Vis DRS of all the fabricated samples (**a**,**b**), EIS spectra of BTO/CTO and Bi-BTO/CTO−X samples (**c**).

Electrochemical impedance spectroscopy (EIS) measurements were used to gain insight into the behavior of photo-induced $h^+$–$e^-$ pairs in BTO/CTO and Bi-BTO/CTO-X photocatalysis [27,28]. The results in Figure 4c show that (1) all samples have only one arc in the AC impedance spectral plane, indicating that the photocatalytic reaction involves only surface charge transfer; and (2) the sample Bi-BTO/CTO-50 has the smallest arc radius on the AC impedance Nyquist diagram. It indicates that the electrons migrate more efficiently in Bi-BTO/CTO-50 with higher visible light utilization efficiency and lower photogenerated charge migration resistance. In other words, the loaded metal bismuth can stimulate the generation and transfer of photogenerated electrons and holes and improve photocatalytic activity. The above results show that Bi-BTO/CTO-50 composites have

good carrier separation and migration performance and high NO removal efficiency under visible light.

### 2.4. Photocatalytic Performance of NO Removal

The photocatalytic activity of the catalysts was evaluated by the oxidation reaction of NO under visible light irradiation and the NO removal rate at the ppb level was used as an evaluation criterion. In the absence of light, NO cannot be oxidized. The NO removal rate of pure metallic monomeric Bi under visible light irradiation is approximately 20%, which can be interpreted as almost no photocatalytic activity, indicating that the amorphous bismuth oxide on the surface of metallic Bi is inactive. Interestingly, the photocatalytic activity of the BTO/CTO-X nanocomposites after loading with Bi, Bi-BTO/CTO-15 and Bi-BTO/CTO-100 showed a slight decrease in activity compared to the pure BTO/CTO samples, and the NO removal rates of Bi-BTO/CTO-25 and Bi-BTO/CTO-50 samples increased to 52.06% and 64%, respectively, (Figure 5a) which were higher than those of the individual BTO/CTO. Among the composite photocatalysts, Bi-BTO/CTO-50 (64%) showed the most superior photocatalytic performance even exceeding other types of similarly structured photocatalysts, such as N-CQDs/CaTiO$_3$ (25%) [29]; and BiOIO$_3$/g-C$_3$N$_4$ (57%) [30].

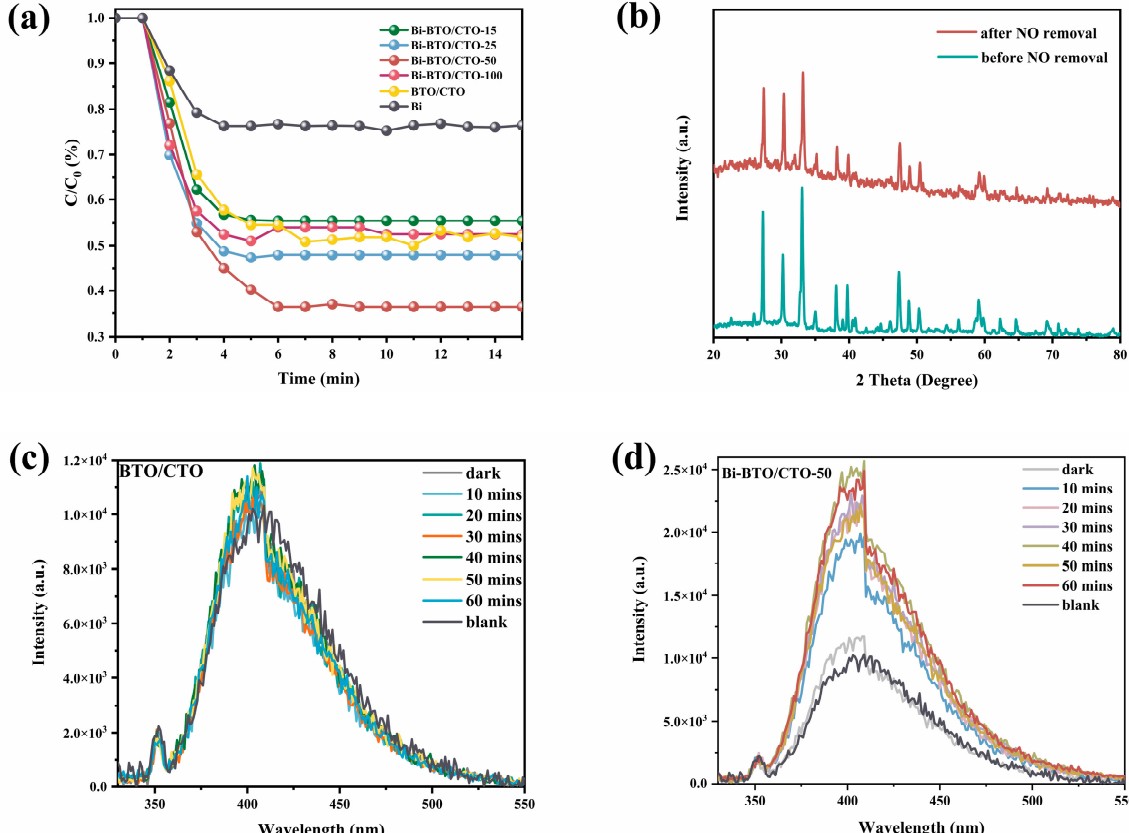

**Figure 5.** Photocatalytic activities of the samples for NO degradation in air under visible light illumination (NO concentration: 600 ppb) (**a**), XRD spectra of the Bi−BTO/CTO−50 before and after the stability test (**b**), The generation of the H$_2$O$_2$ in BTO/CTO system (**c**), The generation of the H$_2$O$_2$ in Bi−BTO/CTO−50 system (**d**).

In practical applications, the persistence and stability of the catalyst is very important [31]. The stability of BTO/CTO and Bi-BTO/CTO-50 was demonstrated by testing the XRD patterns of Bi-BTO/CTO before and after the reaction, and it can be seen that the XRD patterns of Bi-BTO/CTO remained almost unchanged before and after the test (Figure 5b). This result demonstrates that the loading of Bi metal on BTO/CTO nanosheets is a high-

efficiency and durable strategy to significantly improve the photocatalytic performance of BTO/CTO for NO removal.

As shown in Figure 5c, with the prolongation of the illumination time, the content of $H_2O_2$ generated in the BTO/CTO samples did not change with time, while the content of $H_2O_2$ in the Bi-BTO/CTO samples increased first, and then tended to increase with time, further indicating that part of the hydrogen peroxide is decomposed during the light process. Under light conditions, the content of hydroxyl radicals produced in BTO/CTO samples is relatively low. As shown in Figure 5d, in the Bi-BTO/CTO samples, •OH radicals are generated by the decomposition of $H_2O_2$. The results show that $H_2O_2$ and •OH have strong oxidizing properties and can oxidize NO to $NO_3^-$.

The SPR effect is an effect due to the collective oscillation of valence electrons on the surface of metals (Au, Ag, Cu, Al, etc.) under the action of a certain external field (e.g., light). Metal nanoparticles, with the SPR effect, can be controlled by controlling the size, composition and morphology of the metal particles to regulate the light-absorbing properties in the visible–near–infrared region, which is expected to expand the range of their light-harvesting [32]. Metal Bi, a non-precious metal co-catalyst, has received a lot of attention from researchers in recent years due to its very important role in improving the photocatalytic performance of substrate photocatalysts [33]. Because of its low cost and easy availability, can be an excellent potential candidate to replace precious metals (e.g., Au and Pt) for enhancing the photocatalytic performance of base photocatalysts. As in the case of Bi-BTO/CTO-100 (47.4%), the excessive loading of Bi, in turn, reduces the performance of the photocatalytic NO removal. Excessive loading of Bi covers a large portion of the BTO/CTO surface, blocking the irradiation of visible light to the BTO/CTO, which leads to a decrease in the photocatalytic activity of the material.

### 2.5. Possible Photocatalytic Mechanism

The trapping experiments allowed us to determine which reactive radical species are involved in the reaction during the photocatalytic process. As an example, 50 mg of Bi-BTO/CTO-50 sample was used to determine the active species by adding different scavengers. The scavengers p-benzoquinone (PBQ), potassium dichromate ($K_2Cr_2O_7$), tert-butyl alcohol (TBA) and potassium iodide (KI) were chosen to capture $•O_2^-$, $e^-$, •OH and $h^+$ [34,35]. As shown in Figure 6a, when the trapping agent PBQ was added, the conversion of NO under visible light irradiation was inhibited from 64% to 13%, indicating that $•O_2^-$ was the main active substance for NO removal. When TBA was added, the conversion of NO in visible light was similarly inhibited, and the conversion was reduced to about 16%. This indicates that •OH is also the main active substance for NO removal. It is worth noting that, when the scavenger $K_2Cr_2O_7$ was added, it was evident that the conversion of NO was inhibited, from 64% to 12%, and it can be concluded that the $e^-$ is the active species in the oxidation of NO. Meanwhile, after KI was added into the sample and mixed, it was found that the conversion rate of NO was reduced and the photocatalytic efficiency of NO was inhibited, indicating that $h^+$ also plays a very important role in the oxidation of NO in Bi-BTO/CTO heterojunctions and is a non-negligible reactive radical.

As shown in Figure 6b, an ESR signal was detected at approximately g = 2.002, which is attributed to an electron trapped in the OVs [36]. To trap the photoinduced radicals, the DMPO reagent was added to the solution, and no signal was detected under dark conditions, and after 5 min of light, an EPR signal with an intensity ratio of 1:1:1:1 was detected in the Bi-BTO/CTO sample, corresponding to the properties of the DMPO-$•O_2^-$ complex (Figure 6c). Furthermore, as shown in Figure 6d, EPR signals with an intensity ratio of 1:2:2:1, corresponding to the properties of the DMPO-•OH complex, were also detected in the Bi-BTO/CTO sample. This test demonstrates that $•O_2^-$ and •OH are the main active species in the photocatalytic reaction, which is consistent with the results of the capture experiments.

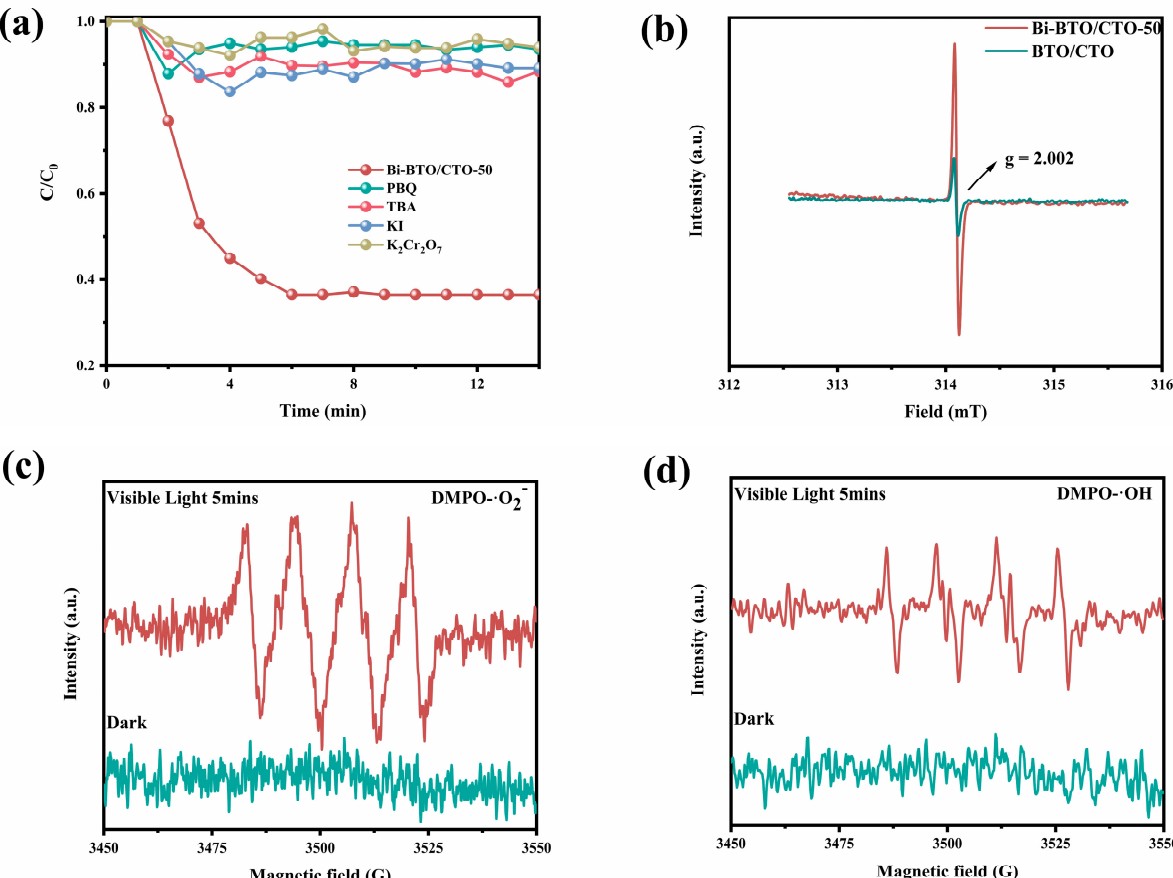

**Figure 6.** Active species trapping of Bi − BTO/CTO − 50 samples (**a**), EPR oxygen vacancy spectrum (**b**), DMPO EPR spin − trapping of Bi − BTO/CTO − 50 for ●$O_2^-$ and ●OH radicals under visible light illumination ($\lambda$ > 420 nm) (**c**,**d**).

Based on the experimental and simulation results, a schematic diagram of the photocatalytic mechanism of NO conversion by Bi−BTO/CTO nanocomposites under visible light irradiation is given in Figure 7. First of all, under solar excitation, photogenerated electrons and holes are produced at the BTO (Equation (1)). The electrons are transferred to the CTO (Equation (2)), and due to the difference between the Fermi energy levels of Bi and the catalyst (−0.17 eV for Bi [37]), the electrons on the CTO are then transferred to a single Bi with a lower work function (Equation (3)); the built-in electric field generated by its SPR effect promotes not only the transfer of photogenerated electrons but also the separation and migration of photogenerated carriers, benefiting the production of ●$O_2^-$ and ●OH radicals leading to NO oxidation [26]. The $h^+$ on the BTO reacts with nitric oxide to form $NO^+$ (Equation (4)), and $O_2$ in the air reacts with $e^-$ to form ●$O_2^-$ (Equation (5)). The superoxide radical continues to react with $NO^+$, $H^+$ to form $NO_2$, $O_2$ and the strongly oxidizing hydrogen peroxide (Equations (6) and (7)), which decomposes under the action of $e^-$ into ●OH and $OH^-$ (Equation (8)), and ●OH react with $NO^+$ to form $NO_3^-$ (Equation (9)). At the same time, the $h^+$ also has strong oxidizing power (Equation (10)) and can oxidize NO in the presence of light to form the end product $NO_3^-$ (Equation (11)).

$$BTO + \text{visible-light} \rightarrow h_{VB}^+ + e_{CB}^- \tag{1}$$

$$e^-(BTO) \rightarrow e^-(CTO) \tag{2}$$

$$e^-(CTO) \rightarrow e^-(Bi) \tag{3}$$

$$h^+ + NO = NO^+ \tag{4}$$

$$e^- + O_2 = \bullet O_2{}^- \tag{5}$$

$$2NO^+ + 2\bullet O_2{}^- = 2NO_2 + O_2 \tag{6}$$

$$\bullet O_2{}^- + 2H^+ + e^- = H_2O_2 \tag{7}$$

$$H_2O_2 + 2e^- = 2\bullet OH + 2OH^- \tag{8}$$

$$\bullet OH + NO^+ = NO_3{}^- + 2H^+ \tag{9}$$

$$2h^+ + NO + H_2O \rightarrow NO_2 + 2H^+ \tag{10}$$

$$NO_2 + h^+ + H_2O \rightarrow NO_3{}^- + 2H^+ \tag{11}$$

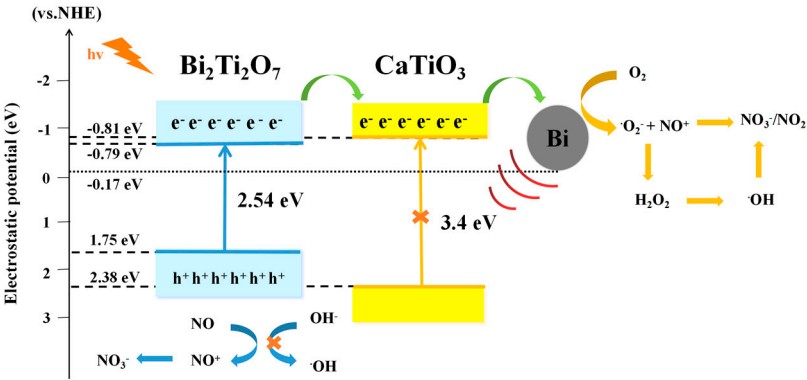

**Figure 7.** The visible-light induced charge separation and proposed photocatalysis mechanism of Bi−BTO/CTO for $NO_x$ purification.

## 3. Experimental Section

### 3.1. Materials and Reagents

All chemicals used are analytically pure and can be used directly without further purification, including KOH, Bi $(NO_3)_3 \cdot 5H_2O$, $HNO_3$, titanium butoxide $Ti(OC_4H_9)_4$, tetrabutyl titanate, ethylene glycol, polyvinyl pyrrolidone and Ca $(NO_3)_2 \cdot 4H_2O$.

### 3.2. Synthesis of Photocatalysts

A one-step hydrothermal method was used to synthesize bismuth titanate/calcium titanate ($Bi_2Ti_2O_7/CaTiO_3$). A total of 56.11 g of KOH was added to a 500 mL beaker, and 500 mL of ultrapure water was added to prepare a 2 mol/L KOH solution. The KOH solution was used for precursor synthesis and the pH of the solution was adjusted before the hydrothermal process. Slowly dropped 0.02 mol of tetrabutyl titanate solution into 300 mL of KOH solution, and then heated the mixture and stirred it for 2–3 h. Then, it was washed with water 3 times and separated by centrifuging to obtain a $TiO(OH)_2$ hydrogel. Transferred the $TiO(OH)_2$ hydrogel to a 250 mL beaker, added 70 mL of deionized water and stirred with a glass rod, and then added 3.78g of $Ca(NO_3)_2 \cdot 4H_2O$ (0.016 mol), 1.94g of $Bi(NO_3)_3 \cdot 5H_2O$ (0.004 mol) and stirred for 30 min, and then adjusted the pH with KOH solution. The pH value was equal to 14. The hydrothermal 160 °C reaction was left for 3 h to obtain $Bi_2Ti_2O_7/CaTiO_3$. The nano $Bi_2Ti_2O_7/CaTiO_3$ was washed 3 times with deionized

water and ethanol, and after vacuum drying, the $Bi_2Ti_2O_7/CaTiO_3$ nanophotocatalyst was obtained. It was then ground into powder and collected in a sample tube as BTO/CTO.

In a typical synthesis, 1 mol/L of nitric acid solution was prepared, 10 mL of nitric acid solution was taken in a clean beaker and 0.364 g of $(BiNO_3)_3 \cdot 5H_2O$ was weighed into the beaker. The mixture was stirred vigorously. After the mixture was dissolved, 55 mL of ethylene glycol was added to it and stirred continuously for 10 min. After a homogeneous solution was formed, 1.34 g of polyvinylpyrrolidone (PVP, M: 24,000) was added and stirring continued until the solution was completely mixed. We waited for the solid to dissolve completely, and then added different masses (1.00, 0.6, 0.3, 0.15 and 0.075 g) of the BTO/CTO sample prepared in the previous step with ultrasonic dispersion for 30 min. It was then hydrothermally incubated at 160 °C for 12 h, washed 3 times with anhydrous ethanol and ultrapure water, respectively, and placed for 12 h in a vacuum oven at 80 °C. The quality ratio was controlled at 0%, 15%, 25%, 50% and 100%, and the corresponding photocatalyst Bi and Bi-BTO/CTO-X (where X represents the mass ratio of Bi to BTO/CTO) were labeled Bi-BTO/CTO-15, 25, 50 and 100. For comparison, the original $Bi_2Ti_2O_7/CaTiO_3$ sample was labeled as BTO/CTO. After it was cooled to room temperature, a gray to black transition product was formed in the hydrothermal kettle. It was then ground into powder and collected in a sample tube.

### 3.3. Photocatalytic NO Removal Performance Test

The photocatalytic experimental setup for photocatalytic removal of nitrogen oxides ($NO_X$) is a continuous flow gas-phase photocatalytic reaction system consisting of a gas supply system, a dynamic gas calibration system, a photocatalytic reactor and a $NO_X$ analysis system. The experiments of photocatalytic removal of NO were carried out in a homemade continuous flow column reactor (R = 5 cm, H = 10 cm). The photocatalytic reactor was made of heat-resistant glass; a total of 0.05 g of sample was weighed and dispersed in 10 mL of ethanol, was sonicated for 20 min and poured in a petri dish to be dried in an oven at 60 °C. The prepared sample disc (R = 3 cm, H = 1.5 cm) was placed in the center of the reactor and a visible light source of a 300 W xenon lamp was incorporated with a 420 nm cut-off filter. The light source was placed vertically just above the quartz reactor and the distance between the bottom of the light source and the sample disc was kept at about 12 cm. Meanwhile, NO gas containing 600 ppb and diluted with cylinder air was continuously passed through the photocatalytic reactor at a gas flow rate of 1 L/min, and the change in NO concentration was continuously measured by an $NO_x$ analyzer. The photocatalytic NO removal efficiency was calculated as follows: $\eta = 1 - C/C_0\%$, where $C_0$ is the initial NO concentration and C is the instantaneous NO concentration.

### 4. Conclusions

In summary, $Bi\text{-}Bi_2Ti_2O_7/CaTiO_3$ was prepared by a mild in situ hydrothermal method. The photocatalytic activity of the metal Bi-loaded materials was significantly higher in visible light compared to the initial $Bi_2Ti_2O_7/CaTiO_3$. When the mass ratio of Bi to BTO/CTO was 50%, the NO removal rate was about 64%. The loading of Bi improves the light absorption capacity and electron transport efficiency of the pristine material $Bi_2Ti_2O_7/CaTiO_3$ from UV to visible light. The built-in electric field effect induced by the SPR effect of the metal Bi promotes not only the transfer of photogenerated electrons but also the segregation and migration of photogenerated carriers, which in turn promotes the production of $\bullet O_2^-$ and $\bullet OH$ radicals leading to the oxidation of NO and ultimately the conversion of NO to $NO_3^-$. This work provides a new strategy for the preparation of efficient environmental photocatalytic composites driven by visible light.

**Author Contributions:** Conceptualization, D.D.; data curation, M.S.; funding acquisition, C.W.; investigation, Q.G.; methodology, Y.Z.; project administration, C.W.; software, D.D.; supervision, C.W.; validation, C.W.; visualization, A.A.A.; writing—original draft, D.D., Q.G. and Y.Z.; writing—review and editing, C.W. and A.R. All authors have read and agreed to the published version of the manuscript.

**Funding:** This work was supported by the National Natural Science Foundation of China (Nos. 52161145409, 21976116). The authors acknowledge Researchers Supporting Project number (RSPD2023R691), King Saud University, Riyadh, Saudi Arabia.

**Data Availability Statement:** Not applicable.

**Conflicts of Interest:** The authors declare no conflict of interest.

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
