# Peer review of "Metal Bi Loaded Bi2Ti2O7/CaTiO3 for Enhanced Photocatalytic Efficiency for NO Removal under Visible Light"

_catalysts, doi:10.3390/catal13081169_

Round 1
Reviewer 1 Report
The present manuscript entitled “Metal Bi loaded Bi2Ti2O7/CaTiO3 for enhanced photocatalytic efficiency for NO removal under visible light” describes the photocatalytic efficiency for NO removal under visible light by using Bi loaded Bi2Ti2O7/CaTiO3. The objective and justification of the work are clear. Therefore, I recommend it for publication. However, some issues are detailed below which need to be addressed before its final acceptance in this journal.
1. In line 50, after Jiang et al [15] replace the full stop with comma. Similarly, in line 55, after Yang et al [16] replace the full stop with comma.
2. In materials and reagents section, check the typo errors in the formulae. Bi (NO3) 3 ·5H2O to “ Bi (NO3)3 ·5H2O”; Ca (NO3) 2 ·4H2O to “Ca (NO3)2 ·4H2O”
3. Titanium butoxide formula was incorrect- correct it.
4. Line 78-80, molarity was not 2 mol/L . calculate once more the molarity or weight of KOH.
5. In the synthesis of photocatalyst, in the initial part, 50 mL KOH solution was prepared, but in the second part author mentioned that, slowly drop 0.02 mol of tetrabutyl titanate solution into 300 mL of KOH solution heat and stir for 2-3 hours. Which one is correct?
6. The author used, “0.016 mol of calcium nitrate tetrahydrate, and 0.004 mol of bismuth nitrate pentahydrate” in synthesis. I recommend the author to write both amount in g and as well as in mol in order to eliminate any errors while writing the manuscript.
7. By considering the above points, author must rewrite the entire synthesis part in the revised manuscript.
8. In line 111, 4 peaks for which compound? Author has to write the compound name clearly.
9. In the XRD written as “The XRD patterns of CaTiO3 and Bi2Ti2O7 samples have been reported in previous articles”. cite that article
10. More discussion needs to be written in xrd part.
11. How the Photocatalytic performance of NO removal were performed. Write the detailed experimental method.
12. In the results and discussion part the author has to compare their results with the published results.
Minor editing of English language required
Author Response
Dear Reviewer,
Thank you for your decision and constructive comments on my manuscript. We have carefully considered the suggestion of reviewer and make some changes. We have tried our best to improve and made some changes in the manuscript.
The reviewer comments are laid out below in Palatino Linotype font and specific concerns have been numbered. Our response is given in Palatino Linotypel font and changes to the manuscript are given in the red text. Revision notes, point-to-point, are given as follows.Please see attachment.

Reviewer 2 Report
This work reports the synthesis, characterization, and photocatalytic assessment of Bi-BTO/CTO samples for NO removal. The authors present the results with proper material characterization, and a possible mechanism is proposed.
However, some issues need to be addressed before the publication:
1. The description of the photocatalytic evaluation conditions must be clearly described in the experimental part.
2. It is not clear if there is a formation of a heterojunction or how are the interfaces between Bi, BTO, and CTO composed. HRTEM analyses are highly recommended for more clarity.
3. As the SPR effect of Bi has an outstanding role in the catalytic activity of these materials, this effect must be deeply explained in the manuscript.
Author Response
Dear Reviewer,
Thank you for your decision and constructive comments on my manuscript. We have carefully considered the suggestion of reviewer and make some changes. We have tried our best to improve and made some changes in the manuscript.
The reviewer comments are laid out below in Palatino Linotype font and specific concerns have been numbered. Our response is given in Palatino Linotypel font and changes to the manuscript are given in the red text. Revision notes, point-to-point, are given as follows.Please see attachment

Reviewer 3 Report
The given paper studies the catalytic efficiency of the synthesized catalyst based on Metal bi loaded Bi2Ti2O7/CaTiO3 on photocatalytic removal of NO under visible light irradiation. The authors established that modification of parent Bi2Ti2O7/CaTiO3 catalyst by metallic Bi significantly increased its photocatalytic activity because of increasing electron-hole pairs conductivity.
The high quality of graphical materials, appropriate research design and clear results can be attributed to the strength of the research.
On the other hand, the experimental methods part must be improved. Particularly, description of the test methods, regimes and equipments must be adequately described. In addition, the section of «Experimental» is recommended to move after «Results and discussions» according to the template of Catalysts.
The following are some point-by-point concerns:
Line 163: the authors mentioned «…..single B nanospheres….», while I could not observe it from Figure 3b. Maybe the authors should replace nanospheres with nanoparticles.
Line 169: the authors argue that elements are uniformly distributed in the sample, indicating the successful synthesis of Bi-BTO/CTO. However, from Figure 3b it is obvious that elements are not uniformly distributed.
Lines 188-189: the Nyquist curve is discussed as an efficiency indicator for electron migration in the composite. The authors have to provide reference for the similar curve and explain it first.
Line 191: The conclusions are not supported and should be further investigated before making a such statements.
Author Response

(The authors gave the same response as above.)

Round 2
Reviewer 2 Report
The comments made to the authors were acceptably revised and corrected, so I recommend the publication of the manuscript in its present form.